# An expression and function analysis of the CXCR4/SDF-1 signalling axis during pituitary gland development

Jose Mario Gonzalez-Meljem[1,2]*, Sarah Ivins[2], Cynthia Lilian Andoniadou[3], Paul Le Tissier[4], Peter Scambler[2], Juan Pedro Martinez-Barbera[2]*

1 Tecnologico de Monterrey, School of Engineering and Sciences, Mexico City, Mexico, 2 Developmental Biology and Cancer Programme, Birth Defects Research Centre, UCL-Great Ormond Street Institute of Child Health, University College London, London, United Kingdom, 3 Division of Craniofacial Development and Stem Cell Biology, King's College London, Guy's Hospital, London, United Kingdom, 4 Centre for Integrative Physiology, University of Edinburgh, Edinburgh, United Kingdom

* jmgonzalezmeljem@tec.mx (JMGM); j.martinez-barbera@ucl.ac.uk (JPMB)

**Data Availability Statement:** All relevant data are within the paper and its Supporting information files.

## Abstract

The chemokine SDF-1 (CXCL12) and its receptor CXCR4 control several processes during embryonic development such as the regulation of stem cell proliferation, differentiation, and migration. However, the role of this pathway in the formation of the pituitary gland is not understood. We sought to characterise the expression patterns of CXCR4, SDF-1 and CXCR7 at different stages of pituitary gland development. Our expression profiling revealed that SDF-1 is expressed in progenitor-rich regions of the pituitary anterior lobe, that CXCR4 and CXCR7 have opposite expression domains and that CXCR4 expression is conserved between mice and human embryos. We then assessed the importance of this signalling pathway in the development and function of the murine pituitary gland through conditional deletion of CXCR4 in embryonic pituitary progenitors. Successful and specific ablation of CXCR4 expression in embryonic pituitary progenitors did not lead to observable embryonic nor postnatal defects but allowed the identification of stromal CXCR4+ cells not derived from HESX1+ progenitors. Further analysis of constitutive SDF-1, CXCR7 and CXCR4 mutants of the pathway indicates that CXCR4 expression in HESX1+ cells and their descendants is not essential for normal pituitary development in mice.

## Introduction

Proper patterning and specification of the pituitary gland primordium, known as Rathke's Pouch (RP) culminates with the correct arrangement and differentiation of three compartments: the anterior lobe (AL), an intermediate lobe (IL) and a posterior lobe (PL). This process depends on spatio-temporally restricted signalling cues provided by morphogens such as WNTs, FGFs, BMPs and SHH [1, 2]. Mutations or dysregulation in these genes and their pathways lead to pituitary-related conditions of great relevance for paediatric clinical research such as: congenital hypopituitarism, septo-optic dysplasia, and pituitary tumours [3].

**Funding:** JPMB is funded by Cancer Research UK, Children's Cancer and Leukaemia Group (www.cclg.org.uk), Children with Cancer UK (www.childrenwithcancer.org.uk), The Brain Tumour Charity (SIGNAL and EVEREST) (www.thebraintumourcharity.org), Great Ormond Street Hospital Children's Charity (www.gosh.org), the Morgan Adams Foundation (www.morganadamsfoundation.org) and National Institute of Health Research Biomedical Research Centre at the Great Ormond Street Hospital for Children NHS Foundation Trust (www.gosh.nhs.uk), and the University College London. The funders had no role in study design, data collection and analysis, decision to publish, or preparation of the manuscript.

**Competing interests:** The authors have declared that no competing interests exist.

In contrast to morphogens, scarce research has addressed the role of chemokine signalling during pituitary development. The most ancient of the CXC- chemokines is CXCL12, commonly known as SDF-1 (Stromal-Derived Factor 1). It has been highly conserved in sequence and function throughout evolution, which prompted the suggestion that it represents the original chemokine in vertebrates. Its receptor, CXCR4, also appeared earlier in evolution than other chemokine receptors and displays the highest level of conservation among them [4, 5]. Binding of SDF-1 to CXCR4 leads to the transcription of genes related to cell migration, survival, and proliferation [6, 7]. Additionally, the alternate SDF-1 receptor, CXCR7, adds extra control over CXCR4 by acting as a scavenger receptor (thus creating SDF-1 gradients) or by forming heterodimers and affecting receptor sensitization [8–11].

CXCR4/SDF-1 signalling is implicated in the development of various organs and tissues. In the developing nervous system, CXCR4 guides proper migration and arrangement of neuron progenitors of the cortex, hippocampus, cerebellum, and spinal cord [12–16]. CXCR4 is also required for the correct migration of cranial neural crest progenitors which contribute to the pharyngeal arches, peripheral ganglia, and cardiac structures [17–19].

In this work, we describe the expression pattern of CXCR4/SDF-1 axis members in the developing pituitary gland of both humans and mice. We also describe genetic approaches assessing the *in vivo* function of CXCR4 during pituitary development, which reveal potential roles for this signalling pathway in stromal cells but not in hormone producing populations or their progenitors.

## Materials and methods

### Mice

The *Hesx1$^{Cre/+}$* line has been previously described and maintained by the host lab (JPMB) for more than 50 generations in a C57BL/6 background [20, 21]. *Cxcr4$^{fl/fl}$* mice (*Cxcr4$^{tm2Yzo}$*) have been previously described and contain loxP sequences flanking exon 2, which represents 98% of the CXCR4 molecule [22]. *Hesx1$^{Cre/+}$* and *Cxcr4$^{fl/fl}$* mice were bred to obtain *Hesx1$^{Cre/+}$*; *Cxcr4$^{fl/+}$* mice, which were then back-crossed with *Cxcr4$^{fl/fl}$* animals to generate *Hesx1$^{Cre/+}$*; *Cxcr4$^{fl/+}$* and *Hesx1$^{Cre/+}$*;*Cxcr4$^{fl/fl}$* mutants. For this study, *Hesx1$^{+/+}$*;*Cxcr4$^{fl/+}$* and *Hesx1$^{+/+}$*; *Cxcr4$^{fl/fl}$* mice and embryos obtained from the same crosses were used as wild type controls. *Cxcr4$^{-/-}$* embryos were generated by crossing *Cxcr4$^{fl/fl}$* with *β-actin$^{Cre/+}$* mice. *Cxcr7$^{-/-}$* (*Cxcr7$^{tm1Litt}$*) and *Cxcl12$^{GFP/GFP}$* (*Cxcl12tm2Tng*) embryos have been previously described [23].

### Ethics approval

All mouse procedures were performed following UK Home Office Animals (Scientific Procedures) Act 1986 and local institutional guidelines (UCL ethical review committee). Use of human samples was carried out under ethical approval 14 LO 2265. Human embryonic samples were provided by the Joint MRC/Wellcome Trust Human Developmental Biology Resource (grant # 099175/Z/12/Z). The HDBR is a Research Ethics Committee (REC) approved and Human Tissue Authority licensed bank. All donors gave written consent before any tissue collection. General information about the donation and consent processes can be found at: https://www.hdbr.org/general-information. The Health Research Authority's Research Ethics Committee approval for the UCL Institute of Child Health can be downloaded at: https://www.hdbr.org/ethical-approvals.

## Mouse sample collection and processing

For embryological studies, female mice were inspected for vaginal plugs the morning after mating set-up. Noon of the day a plug was observed was determined as 0.5 days *post coitum* (dpc) and pregnant females were sacrificed at the appropriate embryological age. All murine samples were collected and further dissected in ice-cold Dulbecco's Modified Eagle's Medium supplemented with 10% Foetal Calf Serum. Samples to be used for immunostaining or *in situ* hybridization (ISH) were washed in phosphate-buffered saline (PBS) and immediately placed in ice-cold freshly made 4% paraformaldehyde, fixed overnight, and posteriorly dehydrated using ethanol gradients. Dehydrated samples were then paraffin-embedded and cut into 6 μm-thick sections. Unless specified, all experiments performed on mouse embryos and pituitaries were performed with a minimum of four biological replicates from at least 2 different litters.

## Immunostaining

Mouse and human paraffin sections were dewaxed in HistoClear (National Diagnostics) and rehydrated using ethanol gradients. Epitope unmasking for all antibodies was conducted in an antigen retrieval unit (BioCare Medical Decloaking Chamber NXGEN) for 2 minutes at 95˚C in 1 M Sodium Citrate pH 6.0 or in 20 mM Tris, 0.65 mM EDTA, 0.005% Tween pH 9.0. Slides were rinsed and permeabilized for 5 minutes in PBT (PBS, 0.1% Triton-X). Blocking was conducted for 1 hour at room temperature in blocking buffer (PBS, 0.1% Triton-X, 0.15% Glycine, 0.2% bovine serum albumin) supplemented with 10% Heat-Inactivated Sheep Serum (HISS). Primary antibodies were then incubated overnight at 4˚C in blocking buffer supplemented with 1% HISS. Primary antibodies, dilutions and antigen retrieval conditions are shown in Table 1. Primary antibodies were detected by immunohistochemistry using biotin-conjugated secondary antibodies (Dako), or by immunofluorescence, with biotinylated secondary antibodies which were then detected with Streptavidin-Alexa-Fluor 555 conjugate (Thermo). For CXCR4/EMCN and CXCR4/GFP double immunostainings, the CXCR4 antibody was detected through biotin/streptavidin-Alexa-Fluor 555 amplification, while the other primary antibody was detected with Alexa-fluor 488 conjugated secondary antibodies. Secondary antibodies were incubated in blocking buffer (1% HISS) for 1 hour at room temperature. Autofluorescence control was conducted by incubation in 0.1% Sudan Black (Sigma) in 70% ethanol for 2 minutes at room temperature. For immunohistochemical stainings, slides were incubated with an Avidin/Biotinylated-Peroxidase Complex (Vector). Chromogenic detection was then

**Table 1. List of primary antibodies used in this study.**

| Target | Clonality | Clone | Company | Catalog number | Dilution | Antigen retrieval buffer |
|---|---|---|---|---|---|---|
| CXCR4 | Monoclonal | UMB-2 | Abcam | ab124824 | 1:200 | Tris-EDTA |
| Endomucin | Monoclonal | V.5C7 | Sant Cruz | SC-53941 | 1:250 | Tris-EDTA |
| FSH | Polyclonal | N/A | DSHB | N/A | 1:1000 | Sodium Citrate |
| GFP | Polyclonal | N/A | Abcam | ab13970 | 1:300 | Tris-EDTA |
| GH | Polyclonal | N/A | DSHB | N/A | 1:1000 | Sodium Citrate |
| GSU | Polyclonal | N/A | DSHB | N/A | 1:500 | Sodium Citrate |
| ACTH | Monoclonal | N/A | Fitzgerald | N/A | 1:1000 | Sodium Citrate |
| PRL | Polyclonal | N/A | DSHB | N/A | 1:1000 | Sodium Citrate |
| Pit1 | Polyclonal | N/A | DSHB | N/A | 1:500 | Tris-EDTA |
| TSH | Polyclonal | N/A | DSHB | N/A | 1:1000 | Sodium Citrate |
| LH | Polyclonal | N/A | DSHB | N/A | 1:500 | Sodium Citrate |
| Tpit | N/A | N/A | Provided by Jacques Drouin | N/A | 1:200 | Tris-EDTA |

conducted by addition of 3,3'-diaminobenzidine (DAB, Vector) for 2–5 minutes and then counterstained with haematoxylin.

## Preparation of antisense riboprobes

*Cxcr4*, *Cxcr7*, *Cxcl12*, *Bmp4*, *Shh*, *Lhx3* and *Fgf10* antisense riboprobes were previously validated [21, 24–27]. Plasmids were digested for linearization using the appropriate enzymes to produce antisense riboprobes and purified using a PCR purification kit (Qiagen). Reaction products were run in agarose gels to confirm complete linearization. RNA transcription was conducted using a nucleotide mix containing Digoxigenin-11-2'-deoxy-uridine-5'-triphosphate (DIG-UTP). Transcription reactions consisted of: 1µg of linearized plasmid DNA, DIG-RNA labelling mix, 10X transcription buffer, RNase inhibitor, and RNA-polymerase (T3, T7 or SP6 depending on the plasmid of origin). Transcription was conducted for 2 hours at 37˚C. 1µl of the reaction product was run on a 1% agarose gel to verify transcription of a product of the appropriate size. Purification of the probe was conducted on CHROMA SPIN 100 columns (Clonetech) according to manufacturer's instructions.

## *In situ* hybridization

ISH was performed as previously described [27]. All solutions were treated overnight with 0.1% di-ethyl-pyrocarbonate (DEPC) in double distilled MiliQ water and subsequently autoclaved. Paraffin sections were de-waxed in HistoClear (National Diagnostics) and re-hydrated using ethanol series. Slides were then fixed for 20 minutes in cold 4% PFA, treated with Proteinase K (20ug/ml) for 8 minutes, re-fixed in 4% PFA and treated with 0.1M triethalolamine and 0.25% acetic anhydride. Slides were then de-hydrated and air-dried. Antisense riboprobes were diluted 1:100 in 50:50 formamide:hybridization buffer (0.3M sodium chloride, 20mM Tris-hydrochloric acid, 5mM EDTA, 10% Dextran sulphate, 1X Denhardt's reagent and RNAse inhibitor). Following overnight incubation at 65˚C, slides were washed in high-stringency conditions in formamide and saline sodium citrate buffer at 65˚C. Slides were then blocked in 0.1M Tris-HCl pH 7.5, 0.15M NaCl solution supplemented with 10% foetal calf serum (FCS) for 1 hour. The detection of DIG- labelled RNA was conducted using a sheep anti-DIG antibody conjugated to alkaline phosphatase at 1:1000 dilution incubated overnight. Subsequently, slides were washed in 0.1M Tris-HCl pH 9.5, 0.15M NaCl, 0.05 MgCl2 buffer. Chromogenic detection of hybridised probes was conducted by adding nitro-blue tetrazolium chloride (NBT, 4.5 µl /ml) and 5-bromo-4-chloro-3-indolyl phosphate (BCIP, 3.5 µl/ml) in 10% polyvinyl alcohol.

## Clonogenic potential assay

Pituitaries were dissected at postnatal day 21 and separated from the PL and incubated for 4 hours at 37˚C in an enzymatic dissociation mix (0.5% w/v Collagenase type 2 (Lorne Laboratories Ltd.), 0.1x Trypsin (Gibco) and 50 µg/ml DNaseI (Worthington) with 2.5 µg/ml Fungizone (Gibco) in Hank's Balanced Salt Solution (HBSS) (Gibco)). Pituitaries were then manually dissociated by pipetting, followed by enzyme inactivation with DMEM (Gibco) and 10% FCS. Cells were then centrifuged and resuspended in pituitary stem cell media (5% FCS, 20 ng/ml bFGF (R&D systems, 233-FB-025), 50 ng/ml cholera toxin (Sigma, C8052), 50 U/ml penicillin and 50 µg/ml streptomycin). Cells were plated in triplicates of 2000, 4000 and 8000 cells per well in 6-well plates. Cells were cultured for 3 days after which colonies were fixed with 4% PFA, counterstained with haematoxylin and counted. The proportion of counted colonies relative to seeded cells was used to estimate total clonogenic cells by multiplying this value by the total number of cells quantified following dissociation of the pituitary. Analysed pituitaries

included: 4 female mutants ($Hesx1^{Cre/+}$;$Cxcr4^{fl/fl}$), and 2 male and 2 female controls ($Hesx1^{+/+}$; $Cxcr4^{fl/fl}$). The sex of the animals has not been found to affect colony formation at these stages [28].

### Hormone content assay

The pituitaries from 2 males and 2 female mutants and matched controls were dissected at postnatal day 21 and used for measurement of total growth hormone (GH) and prolactin (PRL) content. Quantification was conducted by a radioimmunoassay method as previously described [29].

### Imaging, cell counting and statistics

Visualization of immunohistochemical, ISH and H&E stainings was conducted in a Zeiss Axioplan2 microscope and captured with a Zeiss Axiocam HRc colour camera. Immunofluorescent stainings on paraffin sections were visualized with a Leica DMLM widefield microscope and imaged with a CoolSnap monochrome camera or with a Zeiss Axio Observer with a Hamamatsu ORCA camera. Image processing was conducted using Fiji/ImageJ, which included brightness/contrast enhancement and merging of fluorescence channels to produce composite images.

Cell counting was conducted manually using Fiji/ImageJ and a cell counter plugin. At least 3 sections from each pituitary were counted. Sections were selected to ensure that they were sufficiently apart in distance to prevent counting the same cells more than once (surrounding structures such as the brain and the basisphenoid bone were used as reference).

For the quantification of the percentage area covered by EMCN+ cells in pituitary sections, photographs from several different regions per embryo were loaded in Fiji to manually delineate the pituitary tissue (including the posterior lobe and EMCN+ cells directly in contact with the pituitary surface), from which the total area was calculated. Then, the same region of interest was selected in the EMCN fluorescence channel, converted to binary and the signal was detected by thresholding (255, 255) from which its area was determined.

Statistical analysis was conducted using GraphPad Prism in which the statistical significance level was set as 0.05.

## Results

### The expression pattern of CXCR4 in the developing pituitary gland is conserved between mice and humans

It was previously reported that CXCR4 mRNA is expressed at 11.5 dpc. (days *post-coitum*) in the murine pituitary primordium, Rathke's Pouch (RP) [30]. In this study, we sought first to establish a more detailed characterisation of the expression pattern of CXCR4 mRNA and protein throughout pituitary development in mice (Fig 1). This was conducted by *in situ* hybridisation (ISH) and immunostaining using a well validated riboprobe and antibody [27, 31], respectively. Early in pituitary development, at 10.5 dpc, CXCR4 expression localised mainly to the dorso-medial region of Rathke's Pouch, which is the prospective IL (Fig 1A and 1E). At 12.5 dpc, CXCR4 expression was also observed in the IL (Fig 1B and 1F), as well in stromal cells located between the IL and PL. From 15.5 dpc, continuing through 18.5 dpc and postnatally, CXCR4 expression was restricted mostly to IL cells lining RP's cleft (Fig 1C, 1D, 1G and 1H). In the adult pituitary, diffuse CXCR4 protein expression was also observed throughout the AL, in agreement with previous reports [32, 33]. A diffuse signal was observed in the developing PL at all stages but increased in intensity at 15.5 dpc and 18.5 dpc. Interestingly, CXCR4

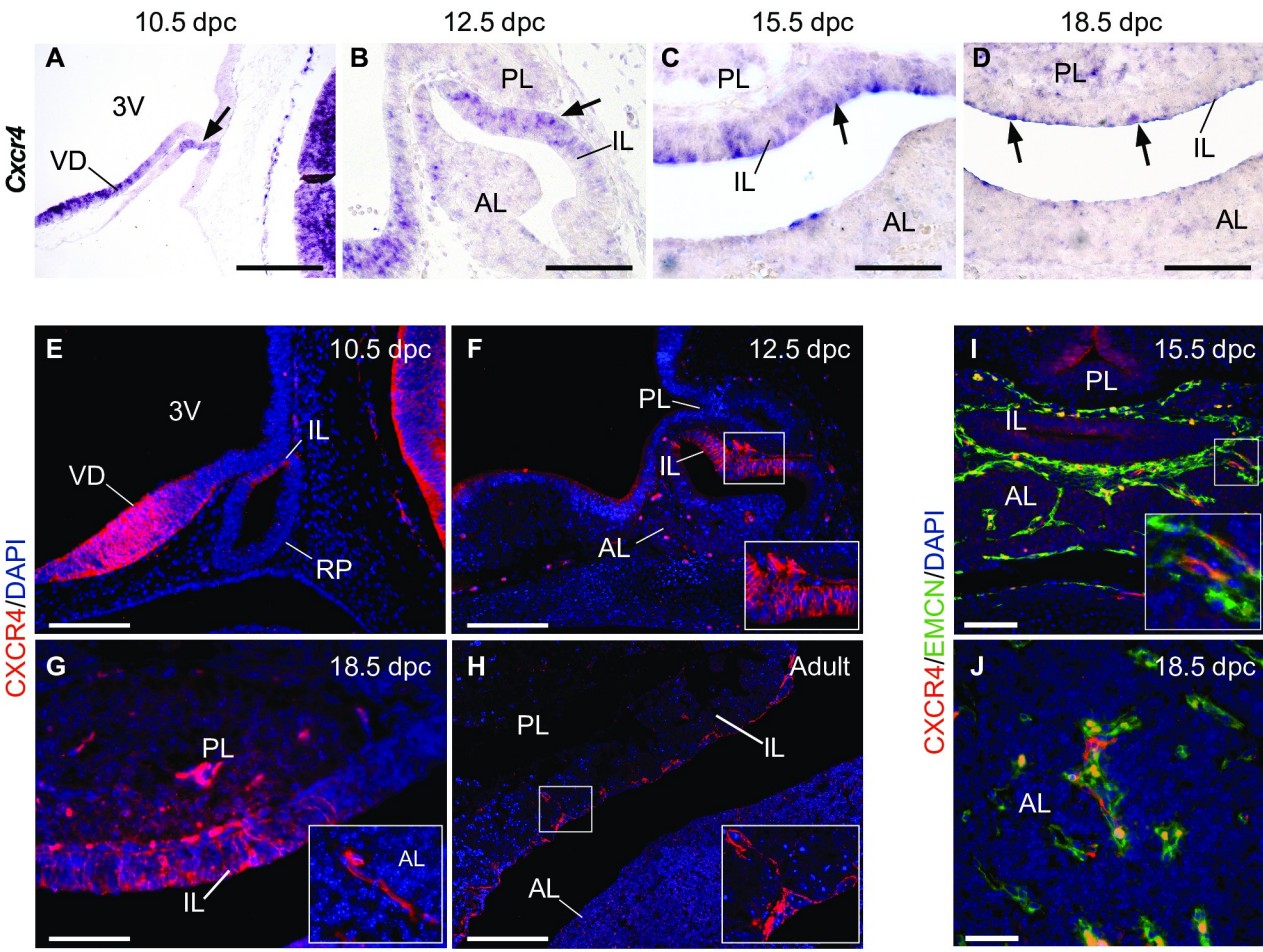

**Fig 1. The expression pattern of CXCR4 mRNA and protein at different stages of murine pituitary development.** (A-D) *In situ* hybridisation (ISH) at 10.5, 12.5, 15.5 and 18.5 dpc shows expression of CXCR4 mRNA in prospective intermediate lobe (IL) of the developing pituitary (arrows). Diffuse positive cells can also be observed in the posterior and anterior lobes (PL and AL, respectively). (E-G) Immunofluorescent staining against CXCR4 protein confirms the ISH expression pattern throughout development. Inset in F: Positive cells are also observed in the boundary between the IL and PL at 12.5 dpc. Inset in G: CXCR4+ cells with elongated morphology are observed in the AL at 18.5 dpc. (H) The IL of adult pituitaries contain CXCR4+ cells lining the pituitary lumen (inset), while a diffuse pattern of CXCR4 expression is observed throughout the AL. (I-J) Double immunostaining at 15.5 dpc and 18.5 dpc for the endothelial marker Endomucin (EMCN) and CXCR4 shows CXCR4+ cells are closely associated to the developing AL vasculature. Counterstain is DAPI. Scale bars: A-F: 200 μm; G-I: 100 μm; J: 50 μm. 3V: Third ventricle; VD: Ventral diencephalon; IL; Intermediate lobe; RP: Rathke's Pouch; AL: Anterior Lobe; PL: Posterior Lobe.

was also observed in cells with elongated morphology of the developing anterior lobe (AL) at 18.5 dpc (Fig 1G). This non-epithelial morphology suggested CXCR4+ cells were related to the pituitary stroma and the developing vasculature, which was corroborated by double immunos-taining for CXCR4 and the endothelial marker Endomucin (EMCN) in the AL of 15.5 dpc and 18.5 dpc pituitaries (Fig 1I and 1J).

We then evaluated the expression pattern of CXCR4 in CS18 and CS20 Carnegie-staged human foetal pituitaries. At both stages, CXCR4 expression closely resembled that of the mouse embryonic pituitary, localising mainly to the prospective IL and posteriorly to RP´s cleft (Fig 2). Positive staining was also observed in the prospective PL. Interestingly, groups of cells with high CXCR4 expression were identified in proximity to the ventral region of the developing pituitary at these stages. These cells displayed filopodial processes and

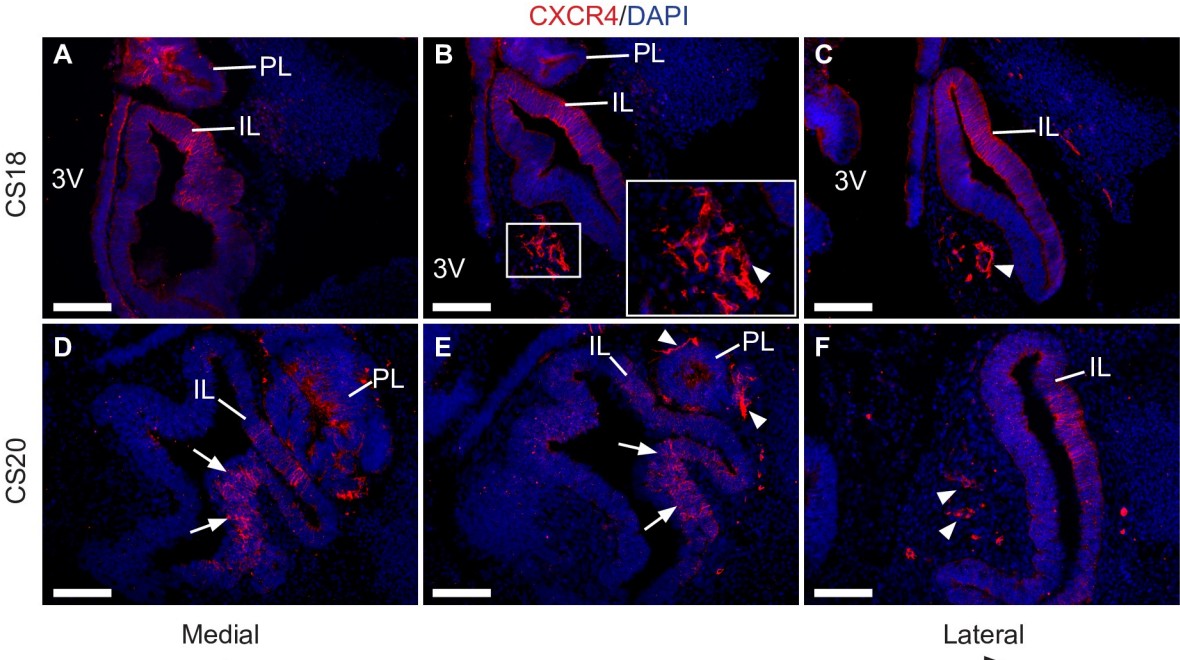

**Fig 2. The expression pattern of CXCR4 at different stages of human embryonic pituitary development.** (A-C) Immunofluorescent staining for CXCR4 at CS18 shows positive cells are found throughout the dorsomedial region of Rathke's Pouch (prospective intermediate lobe). CXCR4+ cells are also observed in the developing infundibulum (prospective posterior lobe). Strong positive signal is observed in cells surrounding luminal structures located ventrally and laterally to the future anterior lobe (arrowheads). (D-F) A similar expression pattern is observed at CS20, with strong CXCR4 expression in the postero-dorsal region of RP (arrows) and in elongated cells located in lateral regions (arrowheads). Counterstain is DAPI. Scale bars: 100 µm. IL; Intermediate lobe; PL: Posterior Lobe; 3V: Third ventricle.

surrounded luminal structures, also suggesting a possible relationship with the future pituitary gland microvasculature. Therefore, the expression pattern of CXCR4 during pituitary development is strikingly similar between mice and humans, implying a conserved function between species.

## The alternative SDF-1 receptor, CXCR7, is expressed in a domain that is complementary to CXCR4

We also evaluated the expression pattern of CXCR7, due to its important role in regulating CXCR4/SDF-1 (Fig 3). While *Cxcr7* and *Cxcr4* were expressed together in the ventral diencephalon (VD) and PL, they displayed mutually exclusive expression patterns in Rathke's Pouch throughout its development. *Cxcr7* mRNA was observed in the oral epithelium and ventral RP regions as early as 10.5 dpc (Fig 3A). At 12.5 dpc and 15.5 dpc, its expression was observed in the PL and AL (Fig 3B and 3C), while low or absent expression levels were observed in the IL. By 18.5 dpc, *Cxcr7* was expressed in cells lining the pituitary cleft in the AL, commonly known as the marginal zone (MZ) (Fig 3D). Interestingly, *Cxcr7* was not expressed in the PL at this stage, in contrast to *Cxcr4* (Fig 1D and 1G). This expression pattern was confirmed at the protein level by immunostaining for CXCR7-GFP in *Cxcr7*$^{GFP/+}$ pituitaries at 12.5 dpc (Fig 3E), while double immunostaining with CXCR4 antibody confirmed mutual exclusivity between CXCR4 and CXCR7 expression domains at this stage (Fig 3F).

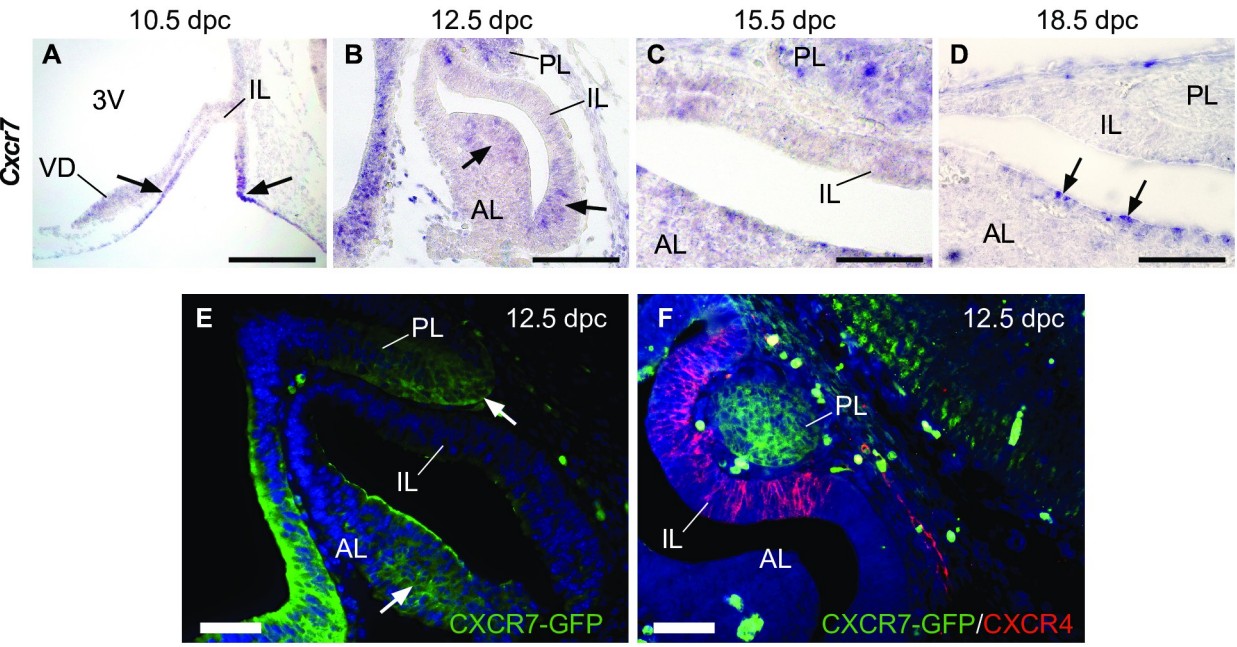

**Fig 3. The expression pattern of CXCR7 during murine pituitary development.** (A) *In situ* hybridisation (ISH) shows *Cxcr7* mRNA is mainly expressed in ventral regions of RP (arrows) at 10.5 dpc, but not in the IL. (B) At 12.5 dpc, *Cxcr7* is expressed in both ventral and posterior regions of the AL lining RP´s cleft (arrows), as well as the PL. (C) At 15.5 dpc, a similar *Cxcr7* expression pattern is observed in the PL and AL. (D) At 18.5 dpc, *Cxcr7* expression is absent in both IL and PL and becomes restricted to cells in the marginal zone of the AL (arrows). (E) Immunofluorescent staining for GFP in 12.5 dpc *Cxcr7*$^{GFP/+}$ embryos at 12.5 dpc shows protein expression in both AL and PL. (F) Double immunostaining for GFP and CXCR4 shows mutually exclusive expression patterns of both proteins. Counterstain is DAPI. Scale bars: A-B: 200 μm; C-D: 100 μm; E-F: 50 μm. 3V: Third Ventricle; VD: Ventral diencephalon; IL; Intermediate lobe; AL: Anterior Lobe; PL: Posterior Lobe.

## SDF-1 is mainly expressed in the mesenchyme and progenitor-rich regions of the anterior lobe

Expression of SDF-1 mRNA was previously reported in mesenchymal tissue surrounding RP at 11.5 dpc [30]. We observed a similar expression pattern at later embryonic stages (Fig 4A and 4B). However, we also observed *Sdf-1* expression distributed throughout the AL at 18.5 dpc, particularly in cells of the MZ (Fig 4B), a region known to contain stem/progenitor cell populations. Immunostaining in SDF-1$^{GFP/+}$ pituitaries also showed SDF-1 protein expression in cells within the AL and MZ at 16.5 dpc (S1A Fig). GFP+ cells were also observed at this stage within the stroma of the anterior-most regions of the AL, at the level of the Median Eminence (ME) (S1B Fig). Therefore, members of the CXCR4/SDF-1 axis display complex expression patterns which may play roles in different cell types and compartments throughout pituitary development.

## CXCR4 is not required in embryonic pituitary progenitors and their progeny for normal development or postnatal function

We assessed the function of CXCR4 signalling in pituitary development by generating *Hesx1*$^{Cre/+}$;*Cxcr4*$^{fl/fl}$ conditional knockout mice and embryos. We chose to use the *Hesx1*$^{Cre/+}$ line to drive conditional recombinase expression as *Hesx1* is one of the earliest expressed genes during RP specification and because most of the pituitary parenchyma is derived from HESX1+ progenitors [34].

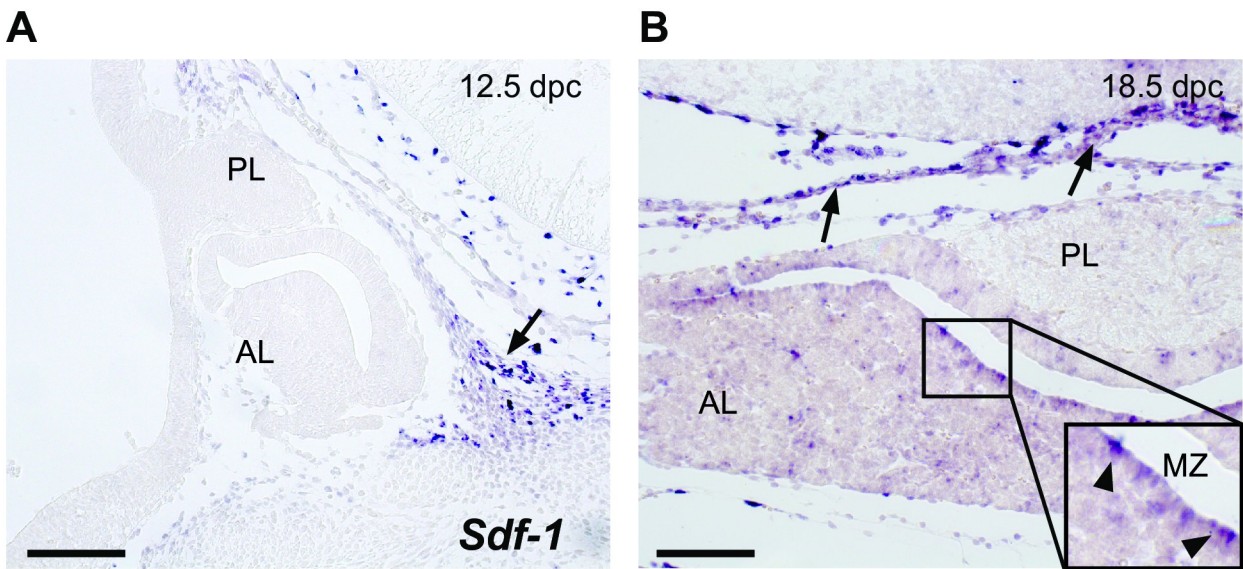

**Fig 4. The expression pattern of *Sdf-1* during murine pituitary development.** (A) *In situ* hybridisation (ISH) at 12.5 dpc shows that SDF-1 mRNA expression is localized in mesenchymal tissue surrounding the developing pituitary gland (arrow). (B) At 18.5 dpc, the supportive mesenchyme still expresses SDF-1 *mRNA* (arrows), although positive cells are also scattered throughout the anterior lobe. Inset: the periluminal region or marginal zone also contains positive cells (arrowheads). Scale bars: 100 μm; AL: Anterior Lobe; PL: Posterior Lobe; MZ: Marginal Zone.

All analysed mutant pituitaries displayed normal morphology and histology (n = 4–6 pituitaries for each developmental stage from 10.5 dpc to 18.5 dpc and adulthood). Successful and specific removal of CXCR4 mRNA and protein expression in *Hesx1^{Cre/+}*;*Cxcr4^{fl/fl}* pituitaries was corroborated by ISH and IHC (Fig 5A and 5B). CXCR4 expression was not visibly affected in other organs and structures such as the brain, while clear absence of CXCR4 staining was observed in the prospective IL of mutant pituitaries at 12.5 and 15.5 dpc (Fig 5A). However, positive cells could still be observed in the stroma located between the IL and PL at both stages. Also, at 15.5 dpc, CXCR4+ cells with elongated morphology were observed in the mutant AL stroma. These results indicate that stromal CXCR4+ cell populations are not derived from the HESX1 lineage in the developing pituitary gland. We also found no changes in the expression domain of *Cxcr7* in mutant pituitaries, suggesting an absence of compensation mechanisms between these two receptors in this context (Fig 5B). Additionally, no changes were observed in the expression domain of genes required for proper pituitary development, such as *Fgf10* in the ventral diencephalon and *Lhx3* in RP (Fig 5C). Finally, ICH for pituitary hormones demonstrated all endocrine cell populations reached normal terminal differentiation in mutants (Fig 5D and S2 Fig).

CXCR4-null mice are characterised by embryonic and perinatal lethality mainly due to vascular defects [35, 36]. Here, Chi-square tests indicated normal Mendelian ratios in both 18.5 dpc embryos (N = 59, P = 0.58) and post-weaning mice (N = 69, P = 0.62) obtained from *Hesx1^{Cre/+}*;*Cxcr4^{fl/+}* and *Cxcr4^{fl/fl}* crosses (Fig 6A). This further demonstrated the pituitary-specific ablation of CXCR4 and indicated its expression is not required for embryonic and perinatal viability. For either sex, the genotype factor did not have a statistically significant effect on the weight of mice at weaning (Fig 6B) (N = 42, two-way ANOVA, P = 0.3871), 8 weeks (Fig 6C) (N = 13, two-way ANOVA, P = 0.6554) or 11 weeks of age (Fig 6D) (N = 12, two-way ANOVA, P = 0.2208). Moreover, all mutant adults that we followed were healthy, fertile, and lived well beyond 1 year of age (n = 6). Similarly, the genotype did not significantly influence

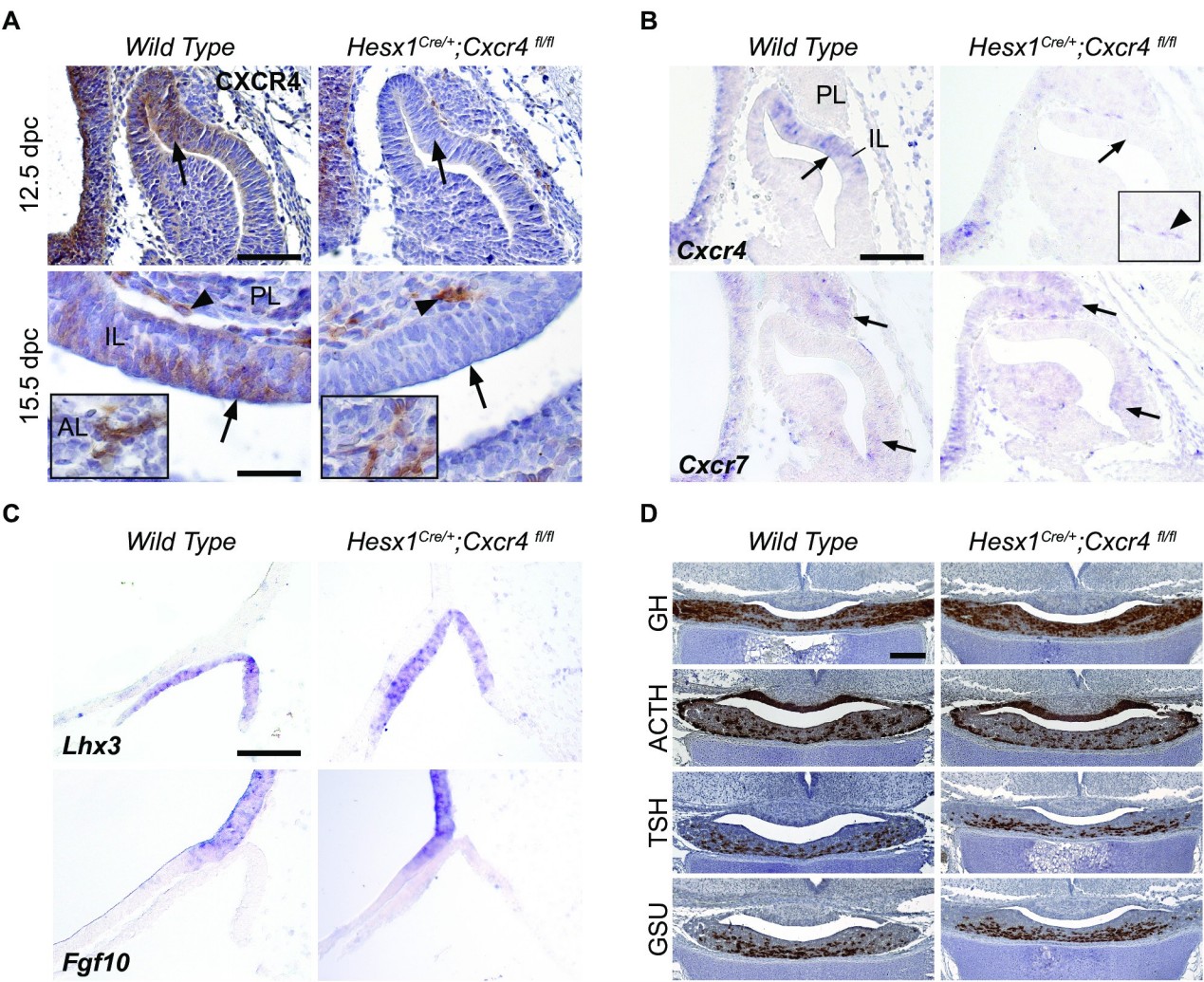

**Fig 5. Molecular characterisation of *Hesx1^Cre/+*;*Cxcr4^fl/fl* pituitaries.** (A) IHC against CXCR4 in wild type and *Hesx1^Cre/+*;*Cxcr4^fl/fl* pituitaries at 12.5 dpc (upper row) and a 15.5 dpc (lower row). Left column: Wild type pituitaries show stereotypical CXCR4 expression IL cells (arrows), but also in cells of the IL/PL boundary and AL (arrowheads, lower row). Right column: *Hesx1^Cre/+*;*Cxcr4^fl/fl* pituitaries show complete absence of CXCR4 in the IL at both stages (arrows). Note that CXCR4+ cells are still present in the IL/PL boundary and in the AL (arrowheads and insets). Haematoxylin counterstain. (B) Upper row: *In situ* hybridisation (ISH) in *Hesx1^Cre/+*;*Cxcr4^fl/fl* and wild type embryos at 12.5 dpc shows absence of *Cxcr4* expression in the IL of mutant pituitaries. Lower row: ISH for *Cxcr7* at 12.5 dpc shows its expression pattern is not affected in *Hesx1^Cre/+*;*Cxcr4^fl/fl* mutants. (C) ISH in 10.5 dpc embryos shows normal expression patterns of *Lhx3* in RP and *Fgf10* in the VD of both wild type and *Hesx1^Cre/+*;*Cxcr4^fl/fl* mutants. (D) ICH in 18.5 dpc embryos for GH, ACTH, TSH and the hormone precursor GSU shows normal terminal differentiation of endocrine populations in *Hesx1^Cre/+*;*Cxcr4^fl/fl* pituitaries. Haematoxylin counterstain. AL: Anterior Lobe; IHC: Immunohistochemistry; ISH: *In Situ* Hybridisation; IL: Intermediate Lobe; PL: Posterior Lobe; GH: Growth Hormone; ACTH: Adrenocorticotropic Hormone; TSH: Thyroid Stimulating Hormone; GSU: Glycoprotein subunit alpha; RP: Rathke´s Pouch; VD: Ventral Diencephalon. Scale bars: A-C: 50 μm; D: 100 μm.

the total pituitary content of growth hormone and prolactin in either males or females (Fig 6E and 6F) (n = 2, two-way ANOVA, *P* = 0.1245 for GH and *P* = 0.6224 for PRL). Because of CXCR4's known involvement in stem cell maintenance, we assayed the effect of CXCR4 knockout in the pituitary stem/progenitor cell population by an *in vitro* assay of colony formation, which showed no significant difference between mutant and wild type pituitaries at postnatal day 21 (Fig 6G) (n = 4, unpaired Student's t-test, *P* = 0.88). Altogether, these results indicate that CXCR4 expression in HESX1+ RP progenitors or their descendants is dispensable for normal pituitary development and overall function under the studied conditions.

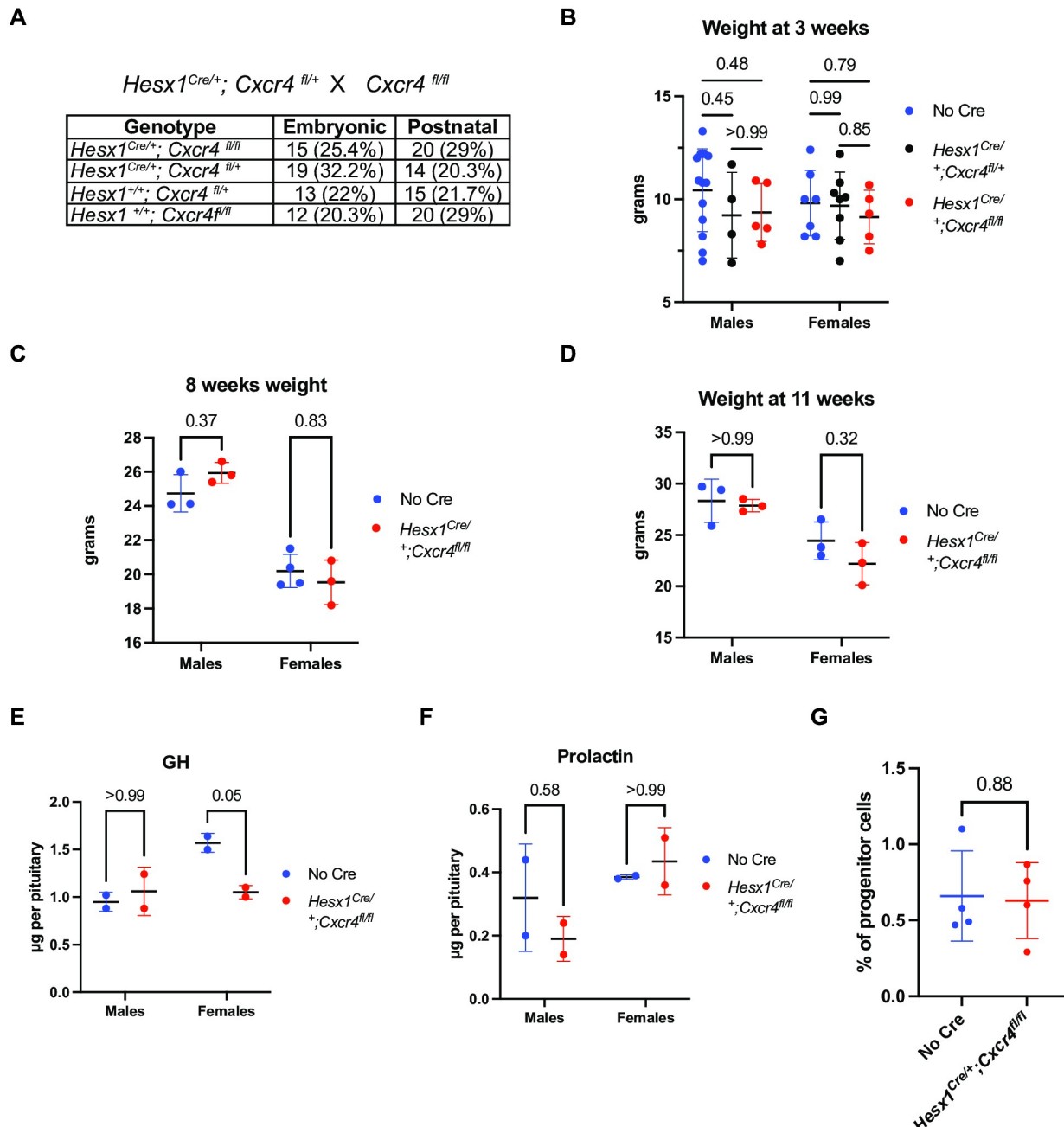

**Fig 6. Phenotypic characterisation of postnatal *Hesx1^{Cre/+}*;*Cxcr4^{fl/fl}* pituitaries and mice.** (A) Table showing normal observed Mendelian ratios for embryos and mice at weaning stage obtained from *Hesx1^{Cre/+}*;*Cxcr4^{fl/+}* and *Cxcr4^{fl/fl}* crosses (N = 69, Chi-square test, p = 0.62). (B-D) Plots showing the weight of *Hesx1^{Cre/+}*;*Cxcr4^{fl/fl}* mice and controls separated by sex at 3, 8 and 11 weeks of age, respectively. Weight is not affected by the genotype at any analysed stage (two-way ANOVA with Bonferroni´s multiple comparisons test, N = 42 and *P* = 0.39 for 3 weeks; N = 13 and *P* = 0.6554 for 8 weeks; N = 12 and *P* = 0.2208 for 11 weeks). (E-F) Plots of total GH and PRL hormone content as measured by radioimmunoassay in male and female pituitaries at postnatal day 21 (n = 2, two-way ANOVA with Bonferroni´s multiple comparisons test, *P* = 0.1245 for GH and *P* = 0.6224 for PRL). (G) Plot of the percentage of progenitor cells present in wild type and mutant pituitaries at postnatal day 21 as evaluated through a clonogenic potential assay (n = 4, unpaired Student's t-test, *P* = 0.88). Horizontal black lines in plots represent mean values and error bars show standard deviations. Significance values are shown for each pair-wise comparison.

We then hypothesized that CXCR4/SDF-1 signalling could play a role in pituitary development by acting on cells of the stroma (i.e., not derived from the HESX1+ lineage) and conducted an analysis of *Cxcr4*[-/-], *Cxcr7*[-/-] and *Sdf-1*[GFP/GFP] (SDF-1 null) embryos. Postnatal assessment of the pituitary phenotype was precluded in these strains because of perinatal lethality. *Cxcr4*[-/-] and *Cxcr7*[-/-] embryos displayed a morphologically normal RP at all observed embryonic stages (S3 Fig). Additionally, *Cxcr4*[-/-] and *Cxcr7*[-/-] embryos showed no changes in the expression of markers for normal pituitary development like Pit1 and TPit, as well for most pituitary hormones (not shown).

SDF-1 null mutants did not display any evident RP phenotype at 11.5 dpc. Interestingly, we observed an aberrant morphology of the developing pituitary at 16.5 dpc and 18.5 dpc in SDF-1 null mutants. This phenotype consisted of a dysmorphic AL and abnormal involutions of the IL, which occasionally fused with the AL in histological sections; as well as structural abnormalities of the basisphenoid bone and presence of ectopic blood vessels (S3F and S3J Fig). Despite these alterations, further analysis of SDF-1 null mutants did not reveal any abnormalities in the expression patterns of *Cxcr4*, *Bmp4*, *Fgf10* and *Lhx3* nor pituitary hormones GH, ACTH, PRL and TSH (S4 Fig). ISH for oxytocin and vasopressin also indicated proper specification of the hypothalamic nuclei in SDF-1 mutants (not shown).

Considering that CXCR4/SDF-1 signalling is important for blood vessel patterning and guidance of endothelial tip cells [37–39], we asked if lack of SDF-1 signalling could proper endothelial cell migration or angiogenesis during formation of the pituitary microvasculature. Staining for EMCN at 15.5 and 16.5 dpc showed that SDF-1 null pituitaries also had a developing endothelial network with EMCN+ cells surrounding luminal structures (S5A and S5B Fig). We then quantified the percentage area occupied by EMCN+ cells in sections from SDF-1 null, SDF-1 wild type and *Hesx1*[Cre/+];*Cxcr4*[fl/fl] pituitaries at 15.5 dpc and found a small, yet statistically significant, reduction in the EMCN+ area in SDF-1 mutants (S5C Fig) (n = 2, nested one-way ANOVA with Holm-Šídák's multiple comparisons test, $P = 0.02$). However, this difference was not maintained at 16.5 dpc (S5D Fig) (n = 2, nested t-test, $P = 0.8$). Due to increased embryonic lethality, we could not analyse in detail SDF-1 null pituitaries at 18.5 dpc or postnatal stages. These results suggest that the SDF-1 null pituitary phenotype is mostly morphological in nature and that it might be secondary to defects in other tissues and structures not related to the pituitary itself, such as products of the surrounding mesenchyme (basisphenoid bone and cranial blood vessels).

## Discussion

In this study, we provide a detailed and extensive expression analysis, at both mRNA and protein levels, of the CXCR4/SDF-1 signalling axis throughout murine pituitary development, which builds on previous early reports [30]. We show that CXCR4 expression is mostly localised to IL cells during development and that IL CXCR4+ cells are also present in adulthood. We also describe a previously unreported population of CXCR4+ cells present within the pituitary stroma which appear to be associated to the developing pituitary vasculature. The identity of this stromal CXCR4+ population remains to be determined, but our observations in *Hesx1*[Cre/+];*Cxcr4*[fl/fl] pituitaries indicate this CXCR4+ population is not derived from the HESX1 lineage. While our analysis of the area covered by EMCN+ cells showed a slight decrease in SDF-1 null pituitaries at 15.5 dpc, we did not observe any differences at 16.5 dpc. This suggests that SDF-1 signals are not required for the initial formation of the endothelial network itself, but questions about its functionality remain to be addressed. Future studies should also evaluate other aspects of blood vessel function such as integrity, permeability, and blood flow. A possibility also remains that SDF-1 is required for the migration of other

non-HESX1 stromal supportive cell types which have been shown to regulate proper micro-vascular function in the pituitary gland [40–42]. Previously, it was shown that conditional knockout of β-catenin in the neural crest leads to morphological defects in the pituitary gland and surrounding structures which greatly resemble that of SDF-1 null mutants. Inter-estingly, in this study it was reported that the pituitary gland displayed a leaky microvascula-ture characterised by the absence of pericytes expressing PDGFRβ [43]. Another study showed that absence of integrin β1 expression in epithelial cells leads to a failure of recruit-ment of pericytes into the developing pituitary gland, which eventually failed completely to form a vascular system [44]. Knowing the importance of CXCR4 in cranial neural crest migration [45], it would be interesting to determine if SDF-1 null pituitaries also display a leaky vasculature with absence of PDGFRβ+ pericytes and if CXCR4 is normally expressed in these cells. A different cell type that could rely on CXCR4 signalling during development are pituitary folliculostellate (FS) cells, a supportive cell type with attributed regulatory and immunosurveillance roles, albeit of unknown origin. In these cells, CXCR4 was shown to mediate their migration and the extension and interconnection of their cytoplasmic pro-cesses [46].

Our analysis of other CXCR4 axis members revealed a complex arrangement of expres-sion domains. SDF-1 was strongly expressed in mesenchymal tissue surrounding the pitui-tary as previously reported. However, we show that SDF-1 is also expressed throughout the AL at later stages of development and notably, its expression also localises to cells of the MZ, which contains a population of pituitary progenitors/stem cells [47]. Further experi-ments should be conducted to determine if SDF-1 is produced by cells expressing stem cell markers (such as SOX2), which could imply a role for SDF-1 in regulating homeostasis and regeneration in the pituitary stem cell niche, as observed in other organs such as the bone marrow and the liver [48, 49]. On the other hand, we show that CXCR4 and CXCR7 display mutually exclusive expression domains throughout pituitary development. We considered this relevant as it has been shown that CXCR7 can act as a scavenger receptor, creating gradients of SDF-1 which fine-tune CXCR4-directed cell migration required for proper development of hypothalamic nuclei, cortex interneurons, the heart and kidney [11, 24, 26, 50, 51].

Our results strongly suggest that CXCR4 is not required in the HESX1 lineage for normal pituitary development. However, the finding that SDF-1 null embryos possess notable mor-phological alterations of the developing pituitary and surrounding tissues, including the mis-placement of large blood vessels, also supports a role for CXCR4 in cells not derived from the HESX1 lineage. The absence of a pituitary phenotype in *Cxcr4*^-/- and *Cxcr7*^-/- embryos, although difficult to reconcile, suggests that these two receptors could have redundant func-tions during the formation of cranial structures surrounding the pituitary gland, such as the basisphenoid bone and large hypophyseal arteries and veins. Further studies should also con-sider recently discovered atypical interactions between CXCR4/CXCR7/SDF-1 axis members with other ligands and receptors [52–54]. Although we could also observe CXCR4+ cells asso-ciated to luminal structures in proximity to the developing pituitary gland in human embryos, further co-expression analyses with endothelial markers such as Endomucin and CD31 will be necessary to unequivocally identify these structures as vascular in nature. This evidence would then support a possible role for CXCR4 in the proper formation or function of the pituitary´s hypophyseal portal system in humans.

Our data also suggests that CXCR4 is not required postnatally in the pituitary gland for sur-vival and viability under basal conditions. However, conclusions regarding its function in the adult pituitary should be made with caution. First, in our long-term viability study, we only maintained mice under basal conditions and second, our pituitary hormone content analysis

was restricted to GH and PRL, also under basal conditions. This leaves open the possibility that CXCR4 might be required for proper pituitary plasticity during stress, injury, or metabolic challenges. This is a feasible hypothesis in view of the remarkable increase in motility and reorganization of specific subsets of somatotrophs after ovariectomy in female mice [55], suggesting a role for chemotactic responses, as well as changes that occur during gestation in the expression patterns of SDF-1 and CXCR4 in somatotrophs and gonadotrophs of the adult ovine pituitary [56].

## Conclusions

In summary, we show that members of the CXCR4/SDF-1 signalling axis are expressed dynamically in different compartments of the developing pituitary gland of mice and humans. Conditional knockout of CXCR4 in murine pituitary embryonic progenitors did not affect its normal development or function under basal conditions, although CXCR4/SDF-1 signalling could still be involved in recruiting supportive stromal cells to the developing vasculature of the pituitary gland.

## Supporting information

**S1 Fig. Immunofluorescent stainings for GFP protein in *Sdf-1*$^{GFP/+}$ pituitaries at 16.5 dpc.** (A) SDF-1-GFP expression is observed in supportive mesenchyme surrounding the pituitary gland (arrow). SDF-1-GFP is expressed within the pituitary anterior lobe (AL), particularly in cells of the marginal zone (MZ) (inset). (B) Anterior-most region of the pituitary gland where SDF-1-GFP is expressed in stromal cells intermingling with the AL (delimited by dotted line). Counterstain is DAPI. Scale bar: 50 μm.
(TIF)

**S2 Fig. Immunohistochemistry for prolactin (PRL), luteinising hormone (LH) and follicle-stimulating hormone (FSH).** Staining in in wild type (top row) and *Hesx1*$^{Cre/+}$;*Cxcr4*$^{fl/fl}$ (bottom row) pituitaries at 18.5 dpc shows normal differentiation of these endocrine populations. Counterstain is haematoxylin. Scale bar: 100 μm.
(TIF)

**S3 Fig. Haematoxylin/Eosin (H&E) staining in wild type, *Sdf-1*$^{-/-}$, *Cxcr4*$^{-/-}$, *Cxcr7*$^{-/-}$ pituitaries at different developmental stages.** (A-D) At 11.5 dpc, the RP from *Sdf-1*$^{-/-}$, *Cxcr4*$^{-/-}$ and *Cxcr7*$^{-/-}$ embryos is not affected. (E) At 16.5 dpc the wild type developing pituitary sits on top of the basisphenoid bone (BB), which is flanked by large blood vessels (arrows). (F) *Sdf-1*$^{-/-}$ pituitaries at this stage have an abnormal shape, invaginations of the intermediate lobe (IL) and a deformed BB. Note ectopic blood vessels intermingling in between the BB (arrows). (G-H) *Cxcr4*$^{-/-}$ and *Cxcr7*$^{-/-}$ pituitaries resemble wild types at 16.5 dpc. (I-J) The *Sdf-1*$^{-/-}$ pituitary phenotype at 18.5 dpc also presents an invaginated IL and deformed BB with ectopic blood vessels (arrows). (K-L) *Cxcr4*$^{-/-}$ and *Cxcr7*$^{-/-}$ pituitaries are morphologically normal at 18.5 dpc. Scale bars: A-D: 100 μm; E-L: 200 μm. RP: Rathke's Pouch; 3V: Third Ventricle; PL: Posterior Lobe; AL: Anterior Lobe; IL: Intermediate Lobe; BB: Basisphenoid Bone.
(TIF)

**S4 Fig. Molecular characterisation of *Sdf-1*$^{-/-}$ developing pituitaries.** (A-B) *In situ* hybridisation in 11.5 dpc embryos shows that *Cxcr4* expression is not significantly altered in *Sdf-1*$^{-/-}$ mutants. (C-F) No differences are found in the expression domains of hypothalamic factors *Bmp4* and *Fgf10*. (G-H) The expression of the transcription factor *Lhx3* (necessary for normal pituitary development) is not altered in *Sdf-1*$^{-/-}$ mutants. (I-P) Fluorescence immunostaining

against different pituitary hormones at 18.5 dpc shows proper terminal differentiation of the pituitary endocrine lineages in *Sdf-1*$^{-/-}$ mutants. DAPI counterstain: K-T. Scale bars: 100 μm. GH: Growth Hormone; ACTH: Adeno-Corticotropic Hormone; TSH: Thyroid Stimulating Hormone.
(TIF)

**S5 Fig. Analysis of the microvasculature in *Sdf-1*$^{-/-}$ developing pituitaries.** (A) Immunofluorescent stainings for the endothelial marker Endomucin (EMCN) in control (left panel) and *Sdf-1*$^{-/-}$ (right panel) pituitaries at 15.5 dpc. (B) EMCN stainings in 16.5 dpc control (left panel) and *Sdf-1*$^{-/-}$ (right panel) pituitaries. Representative images from two biological replicates are shown for each genotype. Insets in all panels show representative close-ups of EMCN+ cells present in luminal structures. DAPI counterstain. Scale bars: 50 μm (10 μm for insets). (C) Quantification of the percentage of area covered by EMCN+ cells in *Sdf-1*$^{+/+}$, *Sdf-1*$^{-/-}$ and *Hesx1*$^{Cre/+}$;*Cxcr4*$^{fl/fl}$ pituitary sections at 15.5 dpc (n = 2, nested one-way ANOVA with Holm-Šídák's multiple comparisons test, $P$ = 0.02). (D) Quantification of the percentage of area covered by EMCN+ cells in *Sdf-1*$^{+/+}$ and *Sdf-1*$^{-/-}$ pituitary sections at 16.5 dpc (n = 2, nested t-test, $P$ = 0.8). Each data point represents a different section. Horizontal black lines in plots represent mean values and error bars show standard deviations. Significance values are shown for each pair-wise comparison.
(TIF)

## Acknowledgments

We thank Dr. Ralf Stumm for kindly donating CXCR7-GFP embryos. We also thank Rukmini K. Reddy for supporting with initial experiments in the characterisation of CXCR4/SDF-1 expression. JMGM belongs to the Molecular and Systems Bioengineering Research Focus Group at Tecnologico de Monterrey.

## Author Contributions

**Conceptualization:** Jose Mario Gonzalez-Meljem, Cynthia Lilian Andoniadou, Juan Pedro Martinez-Barbera.

**Formal analysis:** Jose Mario Gonzalez-Meljem, Juan Pedro Martinez-Barbera.

**Funding acquisition:** Juan Pedro Martinez-Barbera.

**Investigation:** Jose Mario Gonzalez-Meljem, Paul Le Tissier.

**Methodology:** Jose Mario Gonzalez-Meljem, Juan Pedro Martinez-Barbera.

**Project administration:** Juan Pedro Martinez-Barbera.

**Resources:** Sarah Ivins, Peter Scambler.

**Supervision:** Sarah Ivins, Cynthia Lilian Andoniadou, Peter Scambler, Juan Pedro Martinez-Barbera.

**Validation:** Jose Mario Gonzalez-Meljem.

**Visualization:** Jose Mario Gonzalez-Meljem.

**Writing – original draft:** Jose Mario Gonzalez-Meljem.

**Writing – review & editing:** Jose Mario Gonzalez-Meljem, Juan Pedro Martinez-Barbera.

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
