## [Decision Letter · Decision Letter 0]

28 Sep 2022

PONE-D-22-15429An expression and function analysis of the CXCR4/SDF-1 signalling axis during pituitary gland developmentPLOS ONE

Dear Dr. Gonzalez-Meljem,

Thank you for submitting your manuscript to PLOS ONE. After careful consideration, we feel that it has merit but does not fully meet PLOS ONE’s publication criteria as it currently stands. Therefore, we invite you to submit a revised version of the manuscript that addresses the points raised during the review process.

Please address the comments made by reviewer #1 and modify the manuscript.  

We look forward to receiving your revised manuscript.

Kind regards,

Michael Klymkowsky, Ph.D.

Academic Editor

PLOS ONE

Journal Requirements:

2. Please provide additional details regarding participant consent. In the ethics statement in the Methods and online submission information, please ensure that you have specified (1) whether consent was informed and (2) what type you obtained (for instance, written or verbal, and if verbal, how it was documented and witnessed). If the need for consent was waived by the ethics committee, please include this information.

Reviewers' comments:

Reviewer's Responses to Questions

**Comments to the Author**

1. Is the manuscript technically sound, and do the data support the conclusions?

Reviewer #1: Yes

Reviewer #2: Yes

2. Has the statistical analysis been performed appropriately and rigorously? 

Reviewer #1: No

Reviewer #2: Yes

3. Have the authors made all data underlying the findings in their manuscript fully available?

Reviewer #1: Yes

Reviewer #2: Yes

4. Is the manuscript presented in an intelligible fashion and written in standard English?

Reviewer #1: Yes

Reviewer #2: Yes

5. Review Comments to the Author

Reviewer #1: Preliminary/Importance:

Chemokine signalling in pituitary development has not been previously explored in detail. CXCL12 and CXCR4 are amongst the most conserved in sequence and function across species, and have previously been described to play a role migratory cell behaviour during neuronal and neural crest development.CXCR7 is an alternative receptor that regulates CXCL12 dosage. Here Gonzales-Meljem and colleagues report a detailed description of gene expression of members of the CXCR4 pathway in mouse pituitary development, and describe the consequences of loss of function of pathway genes in the gland. Overall the work is of good quality with experiments performed correctly, robustly and with adequate replication. Most of the stated conclusions of the paper were well supported by the experimental evidence presented, with the following exceptions:

“Expression pattern of CXCR4 in the developing pituitary is conserved between mouse and human.” In the mouse colocalization of CXCR4 and Endomucin clearly demonstrates vascular expression of this protein. This is an important discussion point since the authors speculate that the CXCR4 pathway may be activated during postnatal pituitary remodelling. In humans the authors show that there is some CXCR4 expression in the parenchyma consistent with vascular cells, but this point is not supported by a co-expression with Endomucin/CD31, etc to unequivocally identify the vasculature. Inclusion of this experiment would greatly strengthen the paper.

In my opinion, the statement that ‘our analysis of SDF-1 null pituitaries showed a normal pattern of endothelial EMCN+ cells,’ line 375, is too strong a conclusion for the evidence presented in this paper. Firstly, to my eye Shh expression was elevated in the mutants at 11.5 dpc (Figure S4g, H). Secondly, while all of the hormone production cells were generated in the Sdf-1 mutants, the vasculature did not appear entirely normal at 18.5 dpc (Figure S4S, T). To clarify this point more replicates of the Endomucin-stained, matched pituitaries should be shown, ideally at multiple stages with quantification.

The way quantitative data is presented in Figure 6 is unclear and the statistical analysis is flawed.

• The way the animal weight data is shown is very unclear. Should be separated by sex after P21 and mean and SD plotted – alternatively the data could be shown in a table.

• Bar charts should show mean and SD, not SEM, to report the variability in the population.

• The statistical tests on the hormones are invalid since it is impossible to achieve a p<0.05 with a Mann-Whitney U test (i.e., the experiment is underpowered). These data should be removed, reported with no significance attached, or the sample size should be increased.

Scale bars are missing from several figures including; Figure 2, Figure 3E, F, Figure 5B, scale in 5C not defined. Figure S3, Figure S4.

Reviewer #2: This manuscript reports on the investigation of a putative role of the SDF-1/ CXCR4 and CXCR7 signalling system during pituitary gland development. Extensive analyses of expression patterns for this chemokine (SDF-1) and its receptors is provided for both mouse and human developing pituitary gland. The expression patterns are interesting and appeared provocative with particular expression of the CXCR4 receptor in the progenitor niche and intermediate lobe in the developing pituitary. Mouse knockouts (either constitutive or pituitary specific) were used to assess the importance of this signalling pathway for proper pituitary development. Disappointingly, no significant phenotype is observed for any of the mutants or combined mouse mutants.

Despite its negative results, this study is important for the pituitary field as the signalling system has the appearances of an important one. This study will thus be of interest to the field because of its thoroughness.

6. PLOS authors have the option to publish the peer review history of their article (what does this mean?). If published, this will include your full peer review and any attached files.

Reviewer #1: No

Reviewer #2: No

---

## [Author Response · Author response to Decision Letter 0]

22 Nov 2022

Response to reviewers

We are grateful to both reviewers for their time in reading our manuscript and their supportive words. Please find below our responses to reviewer #1´s comments and concerns, which we believe have been extremely useful in improving the quality of the manuscript and presentation of its data.

1. In humans the authors show that there is some CXCR4 expression in the parenchyma consistent with vascular cells, but this point is not supported by a co-expression with Endomucin/CD31, etc to unequivocally identify the vasculature. Inclusion of this experiment would greatly strengthen the paper.

This is a valid point, and we agree that further characterization would be necessary to unequivocally identify these cells as vascular/endothelial in human development. For example, they could be a kind of vascular-associated cell type like pericytes, fibroblasts or some undifferentiated progenitor.

Unfortunately, human fetal samples at those stages are very hard to come by. After a thorough search, we could not find an appropriate sample containing the pituitary region (the head often suffers serious damage during collection) within our group´s tissue samples nor within the Human Developmental Biology Resource tissue bank. We were informed also by the HDBR personnel that new fetal tissue donations were severely hindered during and after the COVID-19 pandemic. 

To address the reviewer´s concern we have changed the following statement in the discussion section:

Although we could also observe CXCR4+ cells associated to luminal structures in proximity to the developing pituitary gland in human embryos, further co-expression analyses with endothelial markers such as Endomucin and CD31 will be necessary to unequivocally identify these structures as vascular in nature. This evidence would then support a possible role for CXCR4 in the proper formation or function of the pituitary´s hypophyseal portal system in humans. 

2. In my opinion, the statement that ‘our analysis of SDF-1 null pituitaries showed a normal pattern of endothelial EMCN+ cells,’ line 375, is too strong a conclusion for the evidence presented in this paper.

We agree with this comment, please see our description of new results addressing Point 4 below. Based on that data, we amended the discussion as follows:

While our analysis of the area covered by EMCN+ cells showed a slight decrease in SDF1 null pituitaries at 15.5 dpc, this difference was not maintained at 16.5 dpc. This suggests that SDF-1 signals are not required for the initial formation of the endothelial network itself, although questions about its functionality remain to be addressed. Future studies should also evaluate other aspects of blood vessel function such as integrity, permeability, and blood flow. A possibility also remains that SDF-1 is required for the migration of other non-HESX1 supportive cell types (e.g. pericytes, pituicytes and folliculostellate cells) which have been shown to be crucial to regulate proper microvascular function in the pituitary gland [41–43]. 

3. Firstly, to my eye Shh expression was elevated in the mutants at 11.5 dpc (Figure S4g, H)

We understand that the reviewer is referring to the strong Shh signal observed in the ventral forebrain region in the 11.5 dpc mutant image (anteriorly to the developing pituitary). We agree that such a strong signal was not observable in the wild-type image and appreciate this was brought to our attention.

However, we stand by our conclusion that the expression pattern of Shh is also not altered in SDF-1 null mutants. The reason for this is that this expression pattern (robust Shh expression in the ventral forebrain region) is very well documented in wild-type embryos at 11.5 dpc. See for example Figure 1I in:

- Carreno, G., Apps, J. R., Lodge, E. J., Panousopoulos, L., Haston, S., Gonzalez-Meljem, J. M., Hahn, H., Andoniadou, C. L., & Martinez-Barbera, J. P. (2017). Hypothalamic sonic hedgehog is required for cell specification and proliferation of LHX3/LHX4 pituitary embryonic precursors. Development (Cambridge), 144(18), 3289–3302. https://doi.org/10.1242/dev.153387

Expression of Shh in the ventral forebrain at 11.5 dpc can also be consulted in the Allen Brain Atlas: https://developingmouse.brain-map.org/experiment/show/100092704 from which we obtained the following image. The arrow indicates the region in question.

We believe that the absence of ventral forebrain Shh signal in our wild-type image is the result of a sectioning artifact derived from incorrect sample alignment, which tilted the embryo in the sagittal plane. This caused that region to no be present in the same sections where the pituitary gland was located. 

Unfortunately, we do not have any unpublished images, stained slides o control samples that we could use to replace the wild-type image for a more representative one. The tilting artifact in our stained wild-type samples means that we do not have sections where both the pituitary and that region are present at the same time. 

We have therefore decided to remove the Shh expression data from supplementary figure S4 and from the main text to avoid showing confusing results. We believe that the unaltered expression of Bmp4, Fgf10 and Lhx3, in addition to all pituitary hormones, is still sufficient evidence of normal expression of factors related to pituitary development.

4. Secondly, while all of the hormone production cells were generated in the Sdf-1 mutants, the vasculature did not appear entirely normal at 18.5 dpc (Figure S4S, T). To clarify this point more replicates of the Endomucin-stained, matched pituitaries should be shown, ideally at multiple stages with quantification.

We agree with the reviewer in that the mutant image an increased signal in the red channel was evident. After close examination of our images, we concluded that a large amount of that signal was derived from unspecific autofluorescence from red blood cells and not from true EMCN+ cells lining blood vessels.

We are attaching here images of a control 18.5 dpc pituitary double-stained for PECAM (green channel) and EMCN (red channel). As it can be observed, a large proportion of the unspecific signal (present in both channels) is produced by anucleate red blood cells. 

When we conducted these previous staining experiments, our lab had not implemented yet any autofluorescence blocking steps in our immunostaining protocols (now described in the Materials and Methods section).

To clarify this data, we conducted new EMCN stainings at 15.5 dpc and 16.5 dpc stages when the pituitary stroma is well defined and there is not yet a large amount of red blood cells circulating within it. We additionally conducted an autofluorescence blocking step with Sudan Black dye which allows us to only observe the staining derived from EMCN cells. We could not replicate stainings at 18.5 dpc due to a lack of samples, as viable null mutants were very rare due to increased embryonic lethality at that stage. 

We quantified the percentage area occupied by EMCN positive signal in several distinct sections of Sdf1-/-, Sdf1+/+ (controls) and Hesx1Cre/+;Cxcr4fl/fl pituitaries at 15.5 dpc. We also quantified at 16.5 dpc with Sdf1-/- and control pituitary sections. Analysis of the data shows a small, yet statistically significant, reduction in the percentage of EMCN+ area in Sdf1-/- pituitaries in comparison to wild-type and Hesx1Cre/+;Cxcr4fl/fl pituitaries at 15.5 dpc. Quantification at 16.5 dpc did not show any statistically differences in the percentage of EMCN+ area.

We have included this data in a new supplementary figure (S5Fig), added a section detailing the quantification procedure in Materials and Methods and amended the results and discussion sections accordingly. 

5. The way the animal weight data is shown is very unclear. Should be separated by sex after P21 and mean and SD plotted – alternatively the data could be shown in a table.

We completely agree with the reviewer and appreciate their effort in improving the quality of our manuscript. We have separated the data in different graphs for each time point and separated by sex with the mean and SD plotted as requested (Figs 6B, 6C, and 6D).

6. Bar charts should show mean and SD, not SEM, to report the variability in the population.

We have changed all charts in Figure 6 to show individual points to facilitate visualization of the data and changed all graphs to show SD instead of SEM.

7. The statistical tests on the hormones are invalid since it is impossible to achieve a p<0.05 with a Mann-Whitney U test (i.e., the experiment is underpowered). These data should be removed, reported with no significance attached, or the sample size should be increased.

Our manuscript indeed indicated that these results were not statistically significant (we stated that p>0.05). To improve the clarity of our results, we changed the text and figure legends to show the exact P values for all statistical tests. We also added the exact calculated P value to the graphs (Figs 6E and 6F). In the case of the hormone content analysis, we had used the Mann-Whitney test as we thought it was more suitable for small sample sizes. To be more consistent with the rest of the analyses, we changed that particular test to a two-way ANOVA with Bonferroni’s correct for multiple comparisons (which still found no significant differences).

8. Scale bars are missing from several figures including; Figure 2, Figure 3E, F, Figure 5B, scale in 5C not defined. Figure S3, Figure S4.

We have added de missing scale bars these figures. We also corrected a few incorrect scale values mentioned in the figure legends of Figs 3&4 and erased some repeated labels in Fig 5 to make it cleaner.

---

## [Decision Letter · Decision Letter 1]

19 Dec 2022

An expression and function analysis of the CXCR4/SDF-1 signalling axis during pituitary gland development

PONE-D-22-15429R1

Dear Dr. Gonzalez-Meljem,

We’re pleased to inform you that your manuscript has been judged scientifically suitable for publication and will be formally accepted for publication once it meets all outstanding technical requirements.

Kind regards,

Michael Klymkowsky, Ph.D.

Academic Editor

PLOS ONE

Additional Editor Comments (optional):

Reviewers' comments:

Reviewer's Responses to Questions

**Comments to the Author**

1. If the authors have adequately addressed your comments raised in a previous round of review and you feel that this manuscript is now acceptable for publication, you may indicate that here to bypass the “Comments to the Author” section, enter your conflict of interest statement in the “Confidential to Editor” section, and submit your "Accept" recommendation.

Reviewer #1: All comments have been addressed

Reviewer #2: All comments have been addressed

2. Is the manuscript technically sound, and do the data support the conclusions?

Reviewer #1: (No Response)

Reviewer #2: Yes

3. Has the statistical analysis been performed appropriately and rigorously? 

Reviewer #1: (No Response)

Reviewer #2: Yes

4. Have the authors made all data underlying the findings in their manuscript fully available?

Reviewer #1: (No Response)

Reviewer #2: Yes

5. Is the manuscript presented in an intelligible fashion and written in standard English?

Reviewer #1: (No Response)

Reviewer #2: Yes

6. Review Comments to the Author

Reviewer #1: (No Response)

Reviewer #2: (No Response)

7. PLOS authors have the option to publish the peer review history of their article (what does this mean?). If published, this will include your full peer review and any attached files.

Reviewer #1: No

Reviewer #2: No

---

## [Editor Report · Acceptance letter]

6 Jan 2023

PONE-D-22-15429R1 

An expression and function analysis of the CXCR4/SDF-1 signalling axis during pituitary gland development 

Dear Dr. Gonzalez-Meljem:

I'm pleased to inform you that your manuscript has been deemed suitable for publication in PLOS ONE. Congratulations! Your manuscript is now with our production department. 

Kind regards, 

on behalf of

Dr. Michael Klymkowsky 

Academic Editor

PLOS ONE